# Continuity of Care and the Quality of Life among Patients with Type 2 Diabetes Mellitus: A Cross-Sectional Study in Taiwan

**DOI:** 10.3390/healthcare8040486

**Published:** 2020-11-14

**Authors:** Pei-Lun Hsieh, Fu-Chi Yang, Yi-Fang Hu, Yi-Wen Chiu, Shu-Yuan Chao, Hsiang-Chu Pai, Hsiao-Mei Chen

**Affiliations:** 1Department of Nursing, College of Health, National Taichung University of Science and Technology, Taichung City 40343, Taiwan; peilun@nutc.edu.tw; 2College of General Education, National Chin-Yi University of Technology, Taichung City 41170, Taiwan; caring686868@gmail.com; 3Institute of Allied Health Sciences, College of Medicine, National Cheng Kung University, Tainan 70101, Taiwan; 4Kuang Tien General Hospital, Taichung, Taichung City 433401, Taiwan; cgh123654@yahoo.com.tw; 5Department of Nursing, Chung Shan Medical University, Taichung City 40201, Taiwan; bethchiu@csmu.edu.tw (Y.-W.C.); pai55215@csmu.edu.tw (H.-C.P.); 6Department of Nursing, Hungkuang University, Taichung City 43302; Taiwan; curie_chao@hotmail.com

**Keywords:** quality of life, continuity of care, disability, type 2 diabetes

## Abstract

Background: Understanding factors associated with the quality of life (QoL) of patients with type 2 diabetes (T2DM) is an important health issue. This study aimed to explore the correlation between continuity of care and quality of life in patients with T2DM and to probe for important explanatory factors affecting quality of life. Methods: This study used a cross-sectional correlation research design. Convenience sampling was adopted to recruit 157 patients, aged 20–80 years and diagnosed with T2DM in the medical ward of a regional hospital in central Taiwan. Results: The overall mean (standard deviation, SD) QOL score was 53.42 (9.48). Hierarchical regression linear analysis showed that age, depression, two variables of potential disability (movement and depression), and the inability to see a specific physician or maintain relational continuity with medical providers were important predictors that could effectively explain 62.0% of the variance of the overall QoL. Conclusions: The relationship between patients and physicians and maintaining relational continuity with the medical providers directly affect patients’ QoL during hospitalization and should be prioritized clinically. Timely interventions should be provided for older adult patients with T2DM, depression, or an inability to exercise to maintain their QoL.

## 1. Introduction

Diabetes mellitus (DM) is a chronic disease that causes serious health and financial burdens globally [1]. Its prevalence rate is 8.5%, affecting approximately 422 million people worldwide [2]. Recently, diabetes care has moved from focusing only on mortality or morbidity [3] to placing more emphasis on quality of life (QoL) [4]. Despite several studies on the disease [4,5,6], there is little research on the correlation between continuity of care (CoC) and QoL among patients with diabetes, and there is a need for in-depth understanding and discussions. Diabetes can cause pathological conditions in multiple organs, including cardiovascular disease, kidney failure, retinopathy, neuropathy, and gastrointestinal lesions [2]; the most common psychosocial outcome is depression [7]. Studies have shown that patients with diabetes are 2.4–4.3 times more likely to develop depression, compared to individuals without the disease. Depression further affects treatment compliance, which has impacts on QoL [8].

Werfalli et al. [4] found that the higher the risk of disability among patients with diabetes, the poorer their QoL. With comorbidities (depression and cardiovascular disease), high body mass index (BMI) or low frequency of physical activity, and cardiac metabolism disorders (hypertension, hyperlipidemia, and high-density lipoprotein levels), the probability of disability among diabetic patients increases [9,10,11]. Increased risk of disability is correlated with poor metabolic control and increased incidence of complications, which affect the QoL of the diabetic patients physiologically, psychologically, socially, and environmentally [11,12]. Therefore, early assessment of the risk of disability is crucial [13].

Disability affects 20–50% of patients with diabetes in the U.S. [14]. The 2015 Global Burden of Disease Study [15] identified diabetes as the ninth leading cause of premature death and disability (1106 disability-adjusted life years (DALYs) for every 100,000 people). The five common domains of disability that affect the QoL of patients are movement, nutrition, cognition, sociability, and depression [4]. Related research has also found that fewer chronic diseases in patients with diabetes was linked with improved overall QoL [4,13], but patients with diabetes had double the likelihood of comorbidities compared with individuals without diabetes [4]. The Charlson comorbidity index (CCI) can predict 1 years’ mortality rates and disease treatment outcomes, which can in turn determine quality-adjusted life years [16].

CoC involves the provision of patient-centered medical care via collaborations between medical providers (physicians, nurses, nutritionists, physical therapists, occupational therapists, and pharmacists) to help patients obtain continuous, consistent, and high-quality medical care services [17,18]. Saint-Pierre et al. [19] highlighted the importance of multidisciplinary (physicians, nurses, and nutritionists) intervention and CoC to patient compliance with treatment. In providing continuity of care, medical providers emphasize on information transmission (diagnosis, symptoms, medication, diets, etc.) and the patient–medical team relationship during hospitalization (patients’ confidence in the care of the medical providers, satisfaction with the emotional support, etc.) in enabling patients to have better QoL, both physically and psychologically.

Haggerty et al. conducted a cross-team survey and identified three aspects of CoC: informational continuity, relational continuity, and management continuity [18]. In other research, CoC was divided into six aspects: relationships with providers in hospital, information transfer to patients, relationship with providers in the community, management of written forms, management of follow-ups, and provider communication management [20,21]. The Patient Continuity of Care Questionnaire (PCCQ) can be used to help professionals understand the nature of continuity of care so they can enable patients with chronic diseases to self-manage their conditions [20]. CoC facilitates reductions in blood glucose, number of re-admissions, number of urgent consultations, comorbidities, and mortality in patients with diabetes [22], improving treatment compliance and illness perception [19].

QoL assessment is an important tool to evaluate the effects of diseases and therapeutic interventions. QoL is related to an individual’s level of self-perceived satisfaction with life [23]. The brief version of the World Health Organization Quality of Life Scale: Brief version (WHOQOL-BREF) has good psychological measurement characteristics, serving as an appropriate tool for cross-cultural assessment. It includes four major categories—social health, psychology, society, and environment—and is used in research to assess how diseases could damage or affect an individual’s subjective well-being in these various fields [24]. Researchers have used the WHOQOL-BREF Taiwan version to assess health-related QoL among diabetic patients [25,26]. Research suggests that type 2 diabetes mellitus (T2DM) may greatly affect QoL in terms of physiological, psychological, social, and environmental well-being [5]. In Taiwan, exploration of the correlation between CoC and QoL has only targeted patients with chronic oral diseases and orthopedic patients [20,27,28]; the purpose of this research was to explore the correlation between CoC and QoL in patients with T2DM and to identify important predictors of QoL.

## 2. Materials and Methods

### 2.1. Study Design and Sample

The study used a cross-sectional descriptive correlational design. Data were collected at the medical ward of a regional hospital in central Taiwan. A questionnaire survey was administered based on convenience sampling, and 157 patients with diabetes were recruited. Criteria for inclusion were age (20–80 years), T2DM diagnosis, and the ability to communicate in Mandarin or Taiwanese. Exclusion criteria were severe dementia and cognitive impairment. Diabetic patients had an average length of stay of 6.88 days. Ninety-eight of the participants (62.4%) had poor blood glucose control (glycated haemoglobin, HbA1c > 8%). The reasons for hospitalization fell into two major categories: diagnosis of another disease and diabetic complications. Patients with diabetes had 2.87 ± 1.39 diseases, including hypertension (86, 36.60%), cardiovascular disease (44, 15.44%), kidney disease (34, 11.93%), cataracts (24, 10.21%), hyperlipidemia (21, 8.94%), and cancer (16, 5.61%). A total of 94 (59.87%) patients had one to two diabetic complications, including pathological changes in the eyes (41, 26.11%), cardiovascular system (33, 21.02%), and kidneys (32, 20.38%).

The questionnaire survey was used for data collection. Before the patients were discharged from the hospital, a structured self-filling questionnaire was administered or face-to-face interviews were conducted, and a researcher and three research assistants were trained in and mutually standardized data collection. Questionnaires were distributed to all patients meeting the inclusion criteria, and the interviewer could selectively give explanations if the patients had questions about items on the questionnaire. For illiterate/literate (self-study) patients (18.5%), the researcher asked the questions item by item in a face-to-face manner to ensure that the patients accurately understood the content of the questionnaire. The questionnaires were collected upon completion to ensure the integrity and quality of the content. The study sample size was estimated using G*Power 3.0.10 (Heinrich-Heine-Universität Düsseldorf, Dusseldorf, Germany) [29] with an alpha value of 0.05, power of 0.8, and effect size set to 0.30. Referring to the effect size of 0.30 suggested by Tavakkoli and Dehghan [6], a sample size of 104 patients was found to be sufficient. In terms of the loss rate, 10% for refusal to participate and 10% for withdrawal from the study were estimated [17]. Data were collected from 1 August 2018 to 31 March 31 2019. 

### 2.2. Measurement

This study used a structural questionnaire. The research tools included the following five major parts.
Sociodemographic characteristics of patients with T2DM included age, gender (male, female), marital status (single, married), living conditions (solitary, non-alone), religious beliefs (without, with), education (illiterate/literate (self-study), primary/junior high/high (vocational) school, junior college and above), employment status (no salary, have salary), and financial situation (categorical variables, main source of income, adequacy of income compared to cost of living). Health status was based on number of diseases, complications, time since diagnosis, type of therapy (oral medicine, insulin therapy, oral medicine plus insulin therapy/oral medicine plus diet therapy), blood glucose status (categorical variables, HbA1c value), frequency of weekly exercise (less than 3 times a week, more than 3 times a week, none), and BMI (categorical variables).Depression status was measured using a shorter version of the Center for Epidemiologic Studies Depression Scale (CES-D) from the Taiwan Longitudinal Study on Aging, including 10 items such as depressed mood, guilt and worthlessness, helplessness and hopelessness, mental or physical retardation, loss of appetite, and sleep disturbances, ranging from 0–30 [30]. CES-D ≥ 10 was defined as having depressive symptoms, and the scale showed good internal consistency and reliability, with a Cronbach’s α value of 0.64–0.87 [31]. The Charlson comorbidity index predicted the 10-year mortality for patients with a range of comorbid conditions, and the calculation was made by forming the sum of 19 medical conditions weighted 1–6, where higher scores indicated a more severe burden of comorbidity [32,33]. The test–retest reliability value of the original scale was > 0.75. The Cronbach’s α value of the internal consistency and reliability of the scale in the study was 0.79.Risk of disability was measured using the assessment scale widely used for care prevention in Japan [34]. The scale comprises 24 true/false questions in the following categories: movement (5 items), nutrition (4 items), cognition (5 items), sociability (5 items), and depression (5 items), with risk for disability in each category indicated by scores ≥ 1 point and the total score ranging from 0 to 24 points. Regarding the risk for disability scale in Chen and Chen’s study, the content validity index value of the expert validity was 0.87 and the Cronbach’s α value was 0.77 for the scale as a whole [35].The Chinese version of the Patient Continuity of Care Questionnaire (PCCQ) used in this study was revised by Chen and Chen for use in Taiwan [20]. There were two subscales and 12 total items in the scale, including relationships with providers in hospital (5 items) and information transfer to patients (7 items). Each item used a 5-point Likert scale, from 1 (strongly disagree) to 5 (strongly agree), and the average score of the sum of each item was calculated as a representative score. The PCCQ showed reliability and validity, and the study indicated that the scale had great concurrent validity and internal consistency [20].The WHOQOL-BREF Taiwan version developed by Professor Kai-Ping Yau was utilized in this study. The QoL measurement comprised four domains including physiological health (7 items), psychological health (6 items), social relationships (4 items), and environmental factors (9 items), and two local questions, for a total of 28 items. Using a 5-point Likert scale, the sum of all the items in each domain was divided by the total number of items in that domain, with the average score of each domain ranging from 1 to 5 points. The representative score of each domain ranged from 4 to 20 points. The overall QoL score was obtained by adding the scores in the four domains, with the total score ranging from 16 to 80 points. The QoL score indicator was the average score/80 × 100, where higher scores indicated better QoL as described in the specific item [36]. The questionnaire was originally used to measure QoL among disabled older adult patients. The content validity index value of the original scale was 0.9, with a Cronbach’s α value of internal consistency of 0.95 [24], and also showed good reliability in the study conducted by Chen and Chen (Cronbach’s α value = 0.93) [20].

### 2.3. Data Collection

All subjects gave their informed consent for inclusion before they participated in the study. The study was conducted in accordance with the Declaration of Helsinki, and the protocol was approved by the Ethics Committee of K Hospital on 12 January 2018 (approval number: REN 10642). All participants recruited for this study provided written informed consent prior to participation. The participants were informed of the aims of the study as well as the benefits, risks, and confidentiality guarantee. The questionnaire survey was then administered to participating patients for approximately 10–15 min.

### 2.4. Statistical Analysis

A statistical analysis was performed using SPSS 25.0 Chinese version (IBM Corp., Armonk, NY, USA). The normality of residuals in the final model was tested using the Kolmogorov–Smirnov test (KS test). Frequency, percentage, mean, and standard deviation were reported for variable description. Independent *t*-test, Pearson correlation, one-way analysis of variance (ANOVA), and hierarchical regression linear analysis were used to determine correlations. For all statistical analyses, a *p*-value of < 0.05 was considered as statistically significant.

## 3. Results

### 3.1. Sociodemographic Characteristics and Health Status of Patients with Diabetes

A total of 163 participants were recruited in this study, and of these, 157 completed questionnaires (attrition rate 3.6%). Among the 157 patients with diabetes, 55.4% were females. The average age was 65.4 ± 12.13 years; 126 (80.25%) had spouses; 144 (97.72%) did not live alone; 103 (65.61%) lived with their spouse; 127 (80.89%) had religious beliefs; and 116 (73.89%) had primary/junior high/senior high (vocational) school education. A total of 41 (26.11%) patients were employed, and 80 (50.96%) reported their main source of income as coming from family members, followed by 44 (28.03%) with income from pension/government grants. Furthermore, 108 (68.80%) patients had sufficient income for their cost of living, while 31 reported inadequate income. Patients with diabetes had 2.87 ± 1.39 diseases, with 88 (56.05%) patients diagnosed with >3 other diseases, including hypertension (86, 36.60%), cardiovascular disease (44, 15.44%), kidney disease (34, 11.93%), cataract (24, 10.21%), hyperlipidemia (21, 8.94%), and cancer (16, 5.61%). The average time since diagnosis for patients with diabetes was 2.94 ± 1.43 years, and time since diagnosis was 1–5 years for 76 patients (48.41%). The average age-adjusted CCI score was 4.91 ± 2.61, with 52 patients (33%) showing a disease load ≥ 6. A total of 61 (38.85%) patients had one complication, including pathological changes in the eyes (41, 26.11%), cardiovascular system (33, 21.02%), and kidneys (32, 20.38%). For therapy, 100 (63.69%) patients took oral medicine and 7 (4.46%) received insulin therapy. In addition, 59 (37.58%) patients had good blood glucose control (HbA1c < 7%); 57 (36.31%) had normal BMI (18 ≥ BMI < 24); 69 (43.95%) had exercised fewer than three times/week (>20 min each) in the last 3 months; and 51 (32.48%) did not exercise. 

### 3.2. QoL, CoC and Risk of Disability among Diabetes Patients

The average QoL questionnaire (WHOQOL–BREF Taiwan version) score among these patients with T2DM was 53.42/80, indicating medium-level QoL (see Table 1). 

The average PCCQ score was 50.11/60, with the mean score for each domain ranging from 4.12 to 4.21. The internal consistency reliability of PCCQ was 0.87. As for internal consistency of each subscale, the Cronbach’s α value of relationships with providers in the hospital was 0.93, while that of information transfer to patients was 0.79. In terms of the domains, the social relationships domain scored the highest on average (13.76 points), followed by the environment domain (13.74), the physiological health domain (13.15), and the psychological health domain (12.76). The higher the score, the better the QoL in the specific domain. Looking at the items, the three with the lowest mean scores were ‘self-consciousness of enjoying life’ (mean, 2.62), ‘overall satisfaction with your health’ (mean, 2.85), and ‘have the opportunity for leisure activities’ (mean, 2.94). The highest mean scores were observed for ‘information transfer to patients’ (4.21 points) and ‘relationships with providers during hospitalization’ (4.12 points). The top three items in the two domains with the lowest scores were ‘medical providers understand your expectations, beliefs and preferences’ (mean of 4.07), ‘medical providers gave emotional support’ (mean of 4.11), and ‘medical providers feel confident of CoC after discharge’ (mean of 4.12). The above results showed that T2DM patients had moderate QoL, and QoL in the psychological category scored the lowest. The results regarding CoC showed that diabetic patients who were about to be discharged felt that their relationship with the medical providers during hospitalization was relatively worse than information transfer. The results showed that medical providers did not fully understand the patients’ expectations, beliefs, and preferences, and there is still room left for improvement by medical providers in terms of providing emotional support and maintaining confidence in providing patients with CoC after their discharge (Table 1).

The study assessed the risk of disability for patients with diabetes across five major categories, including movement, nutrition, cognition, sociability, and depression. The highest mean score was for movement, followed by depression, cognition, sociability, and nutrition, with the mean scores of the risk of disability at 2.03, 1.68, 1.55, 1.38, and 1.38, respectively. Scores ≥ 1 for each aspect indicated risk of disability (Table 1). These results showed that the top three significant potential disabilities for diabetic patients were movement, depression, and cognition. The Cronbach’s α value of the internal consistency and reliability of the scale for this study was 0.83. The Cronbach’s α value of each subscale was 0.80 for movement, 0.65 for nutrition, 0.74 for cognition, 0.85 for sociability, and 0.80 for depression.

### 3.3. Correlation between Sociodemographic Characteristics, Health Status, Risk of Disability, Patient CoC and QoL Among Patients with Diabetes

An analysis of the correlation between the sociodemographic characteristics of patients with diabetes and each QoL domain showed that five variables were significantly correlated with QoL, namely age, number of people living together, education level, employment status (salary), and economic conditions (sufficient income). Age was negatively correlated with the physiological health domain of QoL among patients with diabetes (*r* = −0.204, *p* < 0.01). The number of people living with patients was negatively correlated with QoL; when the number of co-inhabitants was low, all four domains, namely, physiological health (*r* = −0.474, *p* < 0.001), psychological health (*r* = −0.391, *p* < 0.001), social relationships (*r* = −0.240, *p* < 0.01), and environment (*r* = −0.271, *p* < 0.01), showed higher scores. Patients with diabetes and ‘primary/junior high/senior high (vocational)’ or ‘junior college and above’ education had significantly better QoL in the physiological health (*F* = 6.189, *p* < 0.01), psychological health (*F* = 5.675, *p* < 0.01), and environment (*F* = 6.215, *p* < 0.01) domains than those with an ‘illiterate/literate (self-study)’ education level. The four domains of QoL were significantly different among employed patients versus unemployed patients. Employed patients with diabetes had better QoL in the physiological health (*t* = −5.085, *p* < 0.001), psychological health (*t* = −3.854, *p* < 0.001), social relationships (*t* = −2.908, *p* < 0.005), and environment (*t* = −2.035, *p* < 0.005) domains than unemployed patients. The four domains also varied significantly based on income. Those with ‘roughly enough’ had a better QoL than those with ‘slightly insufficient/inadequate’ income, and significant differences were observed in the physiological health (*F* = 3.425, *p* < 0.05), psychological health (*F* = 5.055, *p* < 0.01), and social relationships (*F* = 3.144, *p* < 0.05) domains. Those with ‘sufficient/more than sufficient’ and ‘roughly enough’ income had a better QoL in the environment (*F* = 6.823, *p* < 0.001) domain than those with ‘slightly insufficient/inadequate’ income (Table 2).

The number of diseases, number of complications, time since diagnosis, age-adjusted CCI score, frequency of weekly exercise, and CES-D score were significantly correlated with QoL. Fewer diseases were associated with better QoL in the physiological health (*r* = −0.455, *p* < 0.001), psychological health (*r* = −0.288, *p* < 0.001), social relationships (*r* = −0.208, *p* < 0.01), and environment (*r* = −0.242, *p* < 0.01) domains. Fewer complications were associated with a more positive QoL in the physiological health (*r* = −0.474, *p* < 0.001), psychological health (*r* = −0.391, *p* < 0.001), social relationships (*r* = −0.240, *p* < 0.01) and environment (*r* = −0.271, *p* < 0.001) domains. Time since diabetes diagnosis was negatively correlated with QoL, and a longer time since diagnosis was associated with poorer QoL in the physiological health (*r* = −0.178, *p* < 0.05), psychological health (*r* = −0.181, *p* < 0.05), and environment (*r* = −0.206, *p* < 0.01) domains. Lower age-adjusted CCI scores of patients with diabetes were associated with better QoL in the physiological health (*r* = −0.471, *p* < 0.001), psychological health (*r* = −0.310, *p* < 0.001), social relationships (*r* = −0.191, *p* < 0.05), and environment (*r* = −0.273, *p* < 0.001) domains. Patients who got ‘fewer than three times of exercise a week’ and ‘more than three times of exercise a week’ had better QoL than those who ‘never exercised’, and a positive association was found in the physiological health domain (*F* = 7.316, *p* < 0.001). Those with ‘more than three times of exercise a week’ had better QoL than those who ‘never exercised’ and had a more positive QoL in the psychological health (*F* = 4.752, *p* < 0.01), social relationships (*F* = 4.425, *p* < 0.05), and environment (*F* = 7.402, *p* < 0.001) domains. Lower CES-D scores were associated with a significantly better QoL in the physiological health (*r* = −0.748, *p* < 0.001), psychological health (*r* = −0.675, *p* < 0.001), social relationships (*r* = −0.568, *p* < 0.001), and environment (*r* = −0.540, *p* < 0.001) domains (Table 2).

Two domains of the CoC for patients with diabetes were significantly positively correlated with all of the QoL domains. ‘Patients’ relationships with medical providers’ had a positive correlation with QoL in the physiological health (*r* = 0.212, *p* < 0.01), psychological health (*r* = 0.306, *p* < 0.001), social relationships (*r* = 0.349, *p* < 0.001), and environment (*r* = 0.4366, *p* < 0.001) domains. Greater ‘information transfer to patients’ was associated with better QoL in the physiological health (*r* = 0.288, *p* < 0.001), psychological (*r* = 0.96, *p* < 0.001), social relationships (*r* = 0.404, *p* < 0.05), and environment (*r* = 0.407, *p* < 0.001) domains (Table 2).

Low risk of disability was associated with better QoL in the physiological health (*r* = −0.741, *p* < 0.001), psychological health (*r* = −0.569, *p* < 0.001), social relationships (*r* = −0.450, *p* < 0.001), and environment (*r* = −0.482, *p* < 0.001) domains. The five risk of disability categories (movement, nutrition, cognition, sociability, and depression) and four QoL domains (physiological health, psychological health, social relationships, and environment) reached a statistically significant difference, indicating that fewer risks for disability was associated with a more positive QoL among patients with diabetes. (Table 2).

The abovementioned results showed that the sociodemographic characteristics of the diabetic patients (age, number of people living in household, education level, employment status (salary), economic condition (sufficient income), health status (number of diseases, number of complications, time since diagnosis, age-adjusted CCI score, frequency of weekly exercise, and CES-D score), patient CoC (relationships with providers during hospitalization and information transfer to patients), and risk of disability (movement, nutrition, cognition, sociability, and depression) had a statistically significant difference with the QoL of patients with type 2 diabetes.

### 3.4. Predictors Affecting QoL

To identify factors affecting the QoL of patients with diabetes, a hierarchical linear regression analysis was performed based on the aforementioned results. Variables that had significant correlation with QoL were incorporated into the model in three steps. The first step analyzed patients’ sociodemographic characteristics, including age, number of people living in a household, level of education, economic condition (sufficient income), and employment status (salary); the second step added health status items, which included the number of diseases, number of complications, time since diagnosis, frequency of weekly exercise, age-adjusted CCI score, CES-D score, and the five categories of risk for disability: movement, nutrition, cognition, sociability, and depression; the third step added the two continuity of care domains. Table 3 lists the detailed results.

The WHOQOL-BREF results were normally distributed in the type 2 diabetes groups (Kolmogorov–Smirnov test, *KS* = 0.10, *p* > 0.05). The first step of the analysis showed significant QoL variance based on sufficiency of income and employment (salary), which could effectively explain 10.80% of the overall variance (*F* = 6.955, *p* < 0.001). Income and paid employment could partially predict the QoL of patients with diabetes, whereas adequate income and paid employment were associated with better QoL. The best health status predictor of QoL was CES-D score, which could explain 50.40% of the variance, followed by risk of disability related to movement (5.30%) and age (1.40%); together, they could effectively explain 57.10% of the variance in overall QoL (*F* = 70.087, *p* < 0.001). In the third step, the two CoC domains were included in the regression model, and CES-D remained as the best predictor, explaining as much as 50.40% of the variance, followed by the risk of disability related to movement (5.30%), relationships with provider during hospitalization (4.20%), age (1.10%), and risk of disability related to depression (1.0%). Together, these variables could effectively explain 62.0% of the variance of the overall QoL among patients with diabetes (*F* = 51.811, *p* < 0.001).

## 4. Discussion

The overall mean score of QoL for patients with T2DM was 53.42/80, with an indicator score of 66.7. This QoL score was higher than that reported in other countries [6,37], which may be attributed to differences in patients, ethnicity, and region [7,12].

Shamshirgaran et al. reported an increased degree of functional limitations among older patients with diabetes; and poor vision, impaired hearing, and cognitive dysfunction led to low self-rated QoL [7]. This study found that better QoL was associated with fewer people living with the patients, which could be attributed to simpler interpersonal relationships and better communication with caregivers, reducing the patients’ psychological burdens [38]. Patients in nuclear families had better QoL [5]. The results showed that education level was significantly correlated with QoL in patients with diabetes. A higher education level could be linked with a better understanding of the disease and its impacts as well as better socioeconomic conditions, and better-educated patients were more likely to frequently seek appropriate medical advice and adhere to treatment programs and self-management [4,5,13]. People with paid employment and better financial security were more satisfied with their lives, which could lead to a better perception of QoL [22]. Greater numbers of complications were associated with poorer QoL scores in the physiological health, psychological health, social relationships, and environment domains. Diabetes was characterized by severe complications in its later stages and the deterioration of all domains of QoL [39]. This study also found that longer time of morbidity was associated with a higher number of diseases and a higher comorbidity index, resulting in poorer health conditions in patients with diabetes, which would negatively impact patients’ QoL [4,7,12]. In line with previous findings, patients with diabetes with ‘fewer than three times of exercise a week’ and ‘more than three times of exercise a week’ had better QoL than those ‘without exercise’ [7]. El Assar et al. contended that exercise intervention was the best strategy for patients with diabetes to control blood glucose, and physical activity could help debilitated older adult patients with diabetes to maintain bodily functions and delay disability, leading to better QoL [40].

Factors in the present study included variables related to potential disability, which can be helpful to examine the impact of the risk of disability on the QoL of diabetic patients. The results showed that the five aspects of risk of disability among patients with diabetes were negatively correlated with the domains of QoL, and movement and depression in relation to risk of disability could predict 6.30% of variance in QoL (movement, 5.30%; depression, 1%). These findings were similar to those of previous studies that showed diabetes was strongly associated with physical disability. Nearly half (49%) of the patients with diabetes had a high degree of functional limitations; mobility difficulties increased patients’ dependence on assistance, and diabetic patients’ risk of disability was 2.41 times greater that of individuals without diabetes, showing that risk of disability in movement significantly affects the QoL of patients with diabetes [7,32]. However, other research findings suggested that there was no association between diabetes and disability [7]. Chen and Chen’s research results showed that potential social disability was one of the main factors predicting the QoL of older adults with chronic diseases [35]. However, it was found in the present study that the QoL of diabetic patients was affected more by movement and depression in regard to risk of disability. Risk of disability related to depression could partially predict patients’ QoL. This could be because the average age of the patients with diabetes in this study was 65.24 years, making it more likely that they would focus on their physiological health due to frailty, multiple chronic diseases, and complications, which would increase their psychological burden and risk of depression [10]. Therefore, it is necessary to assess the risk of disability among patients. Clinical strategies to alleviate disabilities (short-term moderately intense physical resistance exercise, blood glucose control, and nutrition education programs) could prevent and manage diabetes, meet the changing needs of patients with diabetes, and promote a support network to improve QoL [4,13].

Another finding in the study was that CoC and overall QoL were positively correlated among patients with diabetes. Better continuity in patients’ relationships with providers during hospitalization and information transfer to patients was linked with better QoL, which was consistent with findings about the use of CoC interventions (discharge preparation services, case management, and a combination of both) to improve QoL among older adult patients with chronic diseases [41,42]. The regression analysis showed that patients’ relationship with providers during hospitalization could predict the QoL of patients with diabetes and explain 4.20% of the variance, and the results showed that the better the relationship with providers during hospitalization, the better the quality of life. The findings were similar to those of previous studies, which found that after discharge, the long-term doctor–patient relationship (interpersonal continuity) reduced mortality and re-admission rate and improved QoL [19,20], but the importance of the ‘relationships with providers during hospitalization’ was more emphasized in this study. Around 25% of patients with diabetes (especially those with comorbidities) may encounter difficulties obtaining CoC, mainly due to the severity of the disease. The inability to see a specific physician or maintain relational continuity with medical providers directly affects patients’ QoL [43]. Improving the QoL of patients with diabetes requires strengthening their CoC, understanding patients’ expectations and care requirements, and providing disease-related information and emotional support to inspire confidence about follow-up care after discharge, achieve effective disease management, and improve disease outcomes [22,41,42].

This study showed that depression was significantly correlated with QoL among patients with diabetes and was the strongest predictor of QoL, explaining 50.40% of the variance. Depression has been shown to be associated with deteriorated health among patients with diabetes. Patients often suffered from dizziness and lower extremity pain, limiting activity functions and cognitive functioning and lowering QoL [13]. Emotional stressors associated with having T2DM have been proven to negatively affect a person’s mental and social well-being [44]. Depression and diabetes distress can reduce control of HbA1c, blood pressure, and cholesterol, treatment adherence, and overall health among patients with type 2 diabetes [45]. Medication adherence, disease severity, and depression were also found to be closely correlated with QoL [44,46]. CoC and providing patients with self-care skills and disease-related information may prevent the risk of disability and development of complications in patients with diabetes. Management of depression in these patients may also efficiently improve patients’ QoL [11,13]. Although the constructed linear regression model had low explanatory power (62.0%) in this study, some variables (age, depression, risk of disability, and CoC) were still found to be influential in the quality of life among patients with type 2 diabetes. Since there are many factors affecting patients with type 2 diabetes and they can vary due to differences in lifestyle, medical environment, drug compliance, and regional difference, other variables should also be included for discussion to find the best linear regression model [7,26,46].

### Study Limitations

The main limitation of the study was the lack of a control group of non-diabetic patients, which would supply a better understanding of the meaning of the results in relation to diabetes, and it is suggested that related data from non-diabetic patients be collected in the future for further research. More than half (58.60%) of the participants in this study were older adult patients and many were illiterate/literate (self-study) (18.5%). There might be errors in the scores obtained due to the use of different methods such as face-to-face interviews or self-completed questionnaires. Consequently, this study was restricted by convenience sampling and the cross-sectional research method because it limited patient recruitment to diabetic patients at the medical ward of a regional hospital in central Taiwan and did not include outpatients or other groups, making it difficult to draw comprehensive inferences. A more rigorous evaluation of the results should be conducted using an appropriate experimental design, which would enable in-depth exploration of the factors influencing QoL among patients with diabetes.

## 5. Conclusions

This study assessed the correlation between CoC and QoL in patients with diabetes and investigated predictors of QoL. Risk of disability related to depression and movement, relationships with providers during hospitalization, and age were the strongest predictors of QoL among patients with diabetes. Therefore, it is recommended that in clinical practice, medical professionals focus on managing depression in older adult patients, detecting depression or psychological distress in these patients at an early stage, and encouraging participation in diabetes support groups and community activities. Furthermore, it is important that medical providers understand the importance of their relationship with patients during hospitalization and provide appropriate CoC to reduce the risk of disability in movement and depression and improve patients’ QoL. In future studies, an expanded sample size is recommended to enable comparison of factors that impact QoL in diabetic and non-diabetic patients, especially regarding potential disability and CoC. The sample can also be divided into younger (20–64 years old) and older (65 years and above) groups to compare differences in CoC and QoL. The research results can also serve as a reference for medical units in the promotion of CoC plans for patients.

## Figures and Tables

**Table 1 healthcare-08-00486-t001:** World Health Organization Quality of Life Scale: Brief (WHOQOL-BREF) Taiwan version, Patient Continuity of Care Questionnaire (PCCQ) scores, and risk of disability in patients with diabetes.

Item	Mean (SD)	Score Indicator	*n* (%)
*WHOQOL–BREF Taiwan version*			
Total QoL score (16–80)	53.42 (9.48)	66.77	
Physiological health domain (4–20)	13.15 (2.87)	65.75	
Psychological health domain (4–20)	12.76 (2.90)	63.8	
Social relationships domain (4–20)	13.76 (2.43)	68.8	
Environment domain (4–20)	13.74 (2.35)	68.7	
*PCCQ*			
Total score of PCCQ (12–60)	50.11 (7.29)	83.52	
Relationships with providers during hospitalization (5–25)	20.61 (3.08)	82.44	
Information transfer to patients (7–35)	29.50 (4.41)	84.28	
*Type 2 diabetes risk of disability (24)*	8.02 (6.13)		
Movement (5)	2.03 (1.75)		113 (72.0)
Nutrition (4)	1.38 (1.16)		114 (72.6)
Cognition (5)	1.55 (1.37)		115 (73.2)
Sociability (5)	1.38 (1.85)		74 (47.1)
Depression (5)	1.68 (1.76)		99 (62.1)

The total score of each quality of life (QoL) domain ranges from 4 to 20 points, where higher scores indicate better QoL for that domain. SD, standard deviation; PCCQ, Patient Continuity of Care Questionnaire. A score of 1 or above in each subscale shows disability risk in that domain.

**Table 2 healthcare-08-00486-t002:** Correlation between sociodemographic characteristics, health status, risk of disability, patient continuity of care, and quality of life among patients with type 2 diabetes (*N* = 157).

Item	Physiological	Psychological	Social	Environment
*Diabetes sociodemographic characteristics*				
Age	−0.204 ***	−0.106	−0.015	−0.083
Number of people living in household	−0.474 ****	−0.391 ***	−0.240 **	−0.271 **
Level of education ^b^	6.189 *****	5.675 **	2.325	6.215 **
Employment status ^a^	−5.085 *** **	−3.854 *** **	−2.908 * **	−2.035 * **
Income ^b^	3.425 *	5.055 **	3.144	6.823 ***
*Health status*
Number of diseases	−0.455 ***	−0.288 ***	−0.208 **	−0.242 **
Number of complications	−0.474 ***	−0.391 ***	−0.240 **	−0.271 ***
Time since diagnosis	−0.178 *	−0.181 *	−0.090	−0.206 **
Age-adjusted CCI score	−0.471 ***	−0.310 ***	−0.191 *	−0.273 ***
Frequency of weekly exercise ^b^	7.316 ***	4.752 **	4.425 *	7.402 **
CES-D (0–30 points)	−0.748 ***	0.675 ***	−0.568 ***	−0.540 ***
*Risk of disability*
Overall	−0.741 ***	−0.569 ***	−0.450 ***	−0.482 ***
Movement	−0.262 ***	−0.476 ***	−0.378 ***	−0.470 ***
Nutrition	−0.483 ***	−0.305 ***	−0.242 **	−0.250 **
Cognition	−0.456 ***	−0.318 ***	−0.225 ***	−0.253 ***
Sociability	−0.624 ***	−0.505 ***	−0.390 ***	−0.418 ***
Depression	−0.635 ***	−0.532 ***	−0.448 ***	−0.415 ***
*PCCQ*
Overall	0.258 **	0.363 ***	0.393 ***	0.405 ***
Relationships with providers during hospitalization	0.288 ***	0.396 ***	0.404 ***	0.407 ***
Information transfer to patients	0.212 **	0.306 ***	0.349 ***	0.366 ***

* *p* < 0.05, ** *p* < 0.01, *** *p* < 0.001; ^a^
*t*-test, ^b^
*F*-test. CCI, Charlson comorbidity index; CES-D, Center for Epidemiologic Studies Depression Scale; PCCQ, Patient Continuity of Care Questionnaire.

**Table 3 healthcare-08-00486-t003:** Correlations among sociodemographic characteristics, health status, risk of disability, continuity of care, and quality of life among patients with type 2 diabetes (*N* = 157).

Item	Step 1 Beta	Adjusted R^2^	Step 2 Beta	Adjusted R^2^	Step 3 Beta	Adjusted R^2^
*Type 2 diabetes sociodemographic characteristics*						
*Age*	−0.044		−0.134 *	0.014	−0.133 *	0.011
*Number of people living in household*	0.036		0.023		0.032	
*Level of education*						
① Primary/junior high school/junior/high school (vocational)	0.066		0.118		0.02	
② Junior college and above	0.0169		0.045		0.06	
③ Illiterate/literate (self-study) (reference group)						
*Income*						
① Sufficient/more than sufficient	0.244 *	0.054	0.071		0.069	0.244 *
② Roughly enough	0.208 *	0.025	0.072		0.084	
③ Slightly insufficient/inadequate						
(reference group)						
*Employment status*	0.297 **	0.029	0.029		0.004	
*Health status*						
Number of diseases			−0.031		−0.019	
Number of complications			−0.049		−0.041	
Time since diagnosis			−0.060		−0.077	
Frequency of weekly exercise						
① <3 times a week			0.021		0.024	
② >3 times a week			0.076		0.062	
③ None (reference group)						
Age-adjusted CCI score			−0.038		−0.019	
CES-D			−0.592 ***	0.504	−0.464 ***	0.504
*Risk of disability*			−0.153		−0.100	
① Movement			−0.314 ***	0.053	−0.263 ***	0.053
② Nutrition			−0.017		−0.040	
③ Cognition			−0.014		−0.052	
④ Social			−0.111		−0.078	
⑤ Depression			−0.136		−0.144 *	0.01
*PCCQ*					0.029	
① Relationships with providers during hospitalization					0.216 ***	0.042
② Information transfer to patients					0.017	
*F* value	6.955 ***		70.087 ***		51.811 ***	
Adjusted *R*^2^ value	0.108		0.571		0.62	

Hierarchical linear regression was used for data analysis. CES-D, Center for Epidemiologic Studies Depression Scale; PCCQ, Patient Continuity of Care Questionnaire. * *p* < 0.05, ** *p* < 0.01, *** *p* < 0.001.

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
