# Peer review of "Continuity of Care and the Quality of Life among Patients with Type 2 Diabetes Mellitus: A Cross-Sectional Study in Taiwan"

_healthcare, 2020, doi:10.3390/healthcare8040486_

Round 1

Reviewer 1 Report

Thank you for your submission of manuscript to Healthcare. I enjoyed reading it because data depicting each patient is extensively done and the finding will improve the clinical practice of diabetic patients. There are several comments.

  1. Recruitment of the study participants is only limited in the hospital ward. That means the results and conclusions are only taken from such patients. Many diabetic patients are seen in outpatient settings, so please add limitation of this study regarding this issue.
  2. You have to specify how many candidates were recruited and how much was the response rate.
  3. CES-D was used to check depressive symptoms and you mention "CES-D ≥ 10 was defined as having depressive symptoms" at Line 141 and 142 in p3. However, you also mention "The average CES-D score was 10.08 ± 1.43, and 74 (47.13%) patients had depression" at Line 215 and 216 in p5. How did you diagnose these patients? Please specify.
  4. Results of Cronbach's α coefficients (or alpha coefficients) in this study should be written in "Results" section. If you use the results of several domains or subscales in each questionnaire, you must mention alpha coefficient of each domain or subscale otherwise you cannot sum up item scores to make them into continuous scales.
  5. When you select factors you mention "a hierarchical linear regression analysis was performed based on the aforementioned results. Variables that had significant correlation with QoL were incorporated into the model in three steps" at Line 315 - 317 in p8. In pages 6-8 there are lots of results of significant correlations. However, it is difficult for readers to follow all the details. Use another table and indicate which factors were selected and how it was done for the final hierarchical linear regression analysis.

Author Response

Dear Editor and Reviewers:
Thank you for your kind letter of November 06, 2020. We find your comments very important. We believe that your suggestions and editing will greatly improve the quality of our paper. We have, hence, revised our manuscript as you advised and marked our corrections in red and underline type and revised the manuscript in accordance with the reviewers’ comments, and carefully proof-read the manuscript to minimize typographical, grammatical, and bibliographical errors.

Reviewer 1
1.Recruitment of the study participants is only limited in the hospital ward. That means the results and conclusions are only taken from such patients. Many diabetic patients are seen in outpatient settings, so please add limitation of this study regarding this issue.

We are grateful for the suggestion. We add one criterion in the Limitations of the Study, as participants are controlled to hospitalized diabetic patients in a regional hospital in Central Taiwan. We do not include outpatients or other groups, making it difficult to present a comprehensive inference. Please see Lines 444-447 on Page 11.

2.You have to specify how many candidates were recruited and how much was the response rate.

Thank you for pointing this out. In total, 163 candidates were recruited; excluding six incomplete questionnaires, the valid response rate is 96.32%. Please see Lines 123-124 on Page 03.

3.CES-D was used to check depressive symptoms and you mention "CES-D ≥ 10 was defined as having depressive symptoms" at Line 141 and 142 in p3. However, you also mention "The average CES-D score was 10.08 ± 1.43, and 74 (47.13%) patients had depression" at Line 215 and 216 in p5. How did you diagnose these patients? Please specify.

Thank you very much for the suggestion. The total score of the 10-item CES-D scale ranges from 0 to 30 points. The self-assessment indicators composed of the statements of depression symptoms are designed as a measure of the depressive symptoms and not for the diagnosis or description of depression. We define those scoring 10 points or higher in the CES-D as “having symptoms of depression”. It is also added in the manuscript that the “CES-D Scale mainly evaluates self-perceived depression symptoms, which are suitable for depression tendency screening and not for diagnosis or description of depression”. In addition, the sentence in the original manuscript, "74 (47.13%) patients had depression" is revised to "74 (47.13%) patients have depressive symptoms. Please refer to Lines 141-143 on Page 3 and Lines 213-214 on Page 05.

4.Results of Cronbach's α coefficients (or alpha coefficients) in this study should be written in "Results" section. If you use the results of several domains or subscales in each questionnaire, you must mention alpha coefficient of each domain or subscale otherwise you cannot sum up item scores to make them into continuous scales.

Thank you very much for the suggestion. (1) We have deleted Cronbach's α coefficients in the Measurement Section, as the Reviewer suggested, and put them in the "Results" section: (1) the Cronbach's α value of the four subscales of QoL is between 0.87-0.93; (2) the Cronbach's α value of the two subscales of PCCQ is between 0.79-0.93; (3) the Cronbach's α value of the five subscales of Risk of disability is between 0.65-0.85. Please refer to Lines 217-220 and 222-224 on Page 5 and Lines 245-248 on Pages 5-6.

5.When you select factors you mention "a hierarchical linear regression analysis was performed based on the aforementioned results. Variables that had significant correlation with QoL were incorporated into the model in three steps" at Line 315 - 317 in p8. In pages 6-8 there are lots of results of significant correlations. However, it is difficult for readers to follow all the details. Use another table and indicate which factors were selected and how it was done for the final hierarchical linear regression analysis.

We appreciate this suggestion. We make the revision of Table 3 in accordance with the Reviewer’s suggestion to allow readers to understand the situation of using the three steps of hierarchical linear regression analysis and to see the final linear regression model from the third step of the hierarchical linear regression analysis. Please see Lines 328-334 and Table 3 on Pages 8-9.

Thank you and all the reviewers for the kind advice.
Sincerely yours,

Hsiao-Mei Chen, RN, PHD

Reviewer 2 Report

Thank you for giving me the opportunity to read your work, which I think is of very high quality. I have a suggestion for the authors: Actually the values of the variance explained by the linear regression models are low, so it seems that the models are not quite adequate. While it is true that this could be due to the sample, I think the authors should discuss the validity of their model in the discussion section.

Author Response

Dear Editor and Reviewers:
Thank you for your kind letter of November 06, 2020. We find your comments very important. We believe that your suggestions and editing will greatly improve the quality of our paper. We have, hence, revised our manuscript as you advised and marked our corrections in red and underline type and revised the manuscript in accordance with the reviewers’ comments, and carefully proof-read the manuscript to minimize typographical, grammatical, and bibliographical errors.

Reviewer 2
1. Thank you for giving me the opportunity to read your work, which I think is of very high quality. I have a suggestion for the authors: Actually the values of the variance explained by the linear regression models are low, so it seems that the models are not quite adequate. While it is true that this could be due to the sample, I think the authors should discuss the validity of their model in the discussion section.

Thank you very much for the comment. According to the Reviewer’s suggestion, due to the low variance being explained by the linear regression model, we discuss the effectiveness of the model in the Discussion section. Please see Lines 432-437 on Page 11.

Thank you and all the reviewers for the kind advice.
Sincerely yours,

Hsiao-Mei Chen, RN, PHD
